# "We are invisible to them"—Identifying the most vulnerable groups in humanitarian crises during the COVID-19 pandemic: The case of Rohingyas and the Host communities of Cox's Bazar

**Rafia Sultana**[ID]\*, **Ateeb Ahmad Parray**[ID], **Muhammad Riaz Hossain**[ID], **Bachera Aktar**[ID], **Sabina Faiz Rashid**[ID]

The Center of Excellence for Gender, Sexual and Reproductive Health and Rights, BRAC James P Grant School of Public Health, BRAC University, Dhaka, Bangladesh

\* rafia.sultana@bracu.ac.bd

**Data Availability Statement:** The available data includes illustrative quotes drawn from the

## Abstract

The COVID-19 pandemic has had an adverse impact on the Rohingya and the Bangladeshi host communities, which have been well documented in the literature. However, the specific groups of people rendered most vulnerable and marginalized during the pandemic have not been studied comprehensively. This paper draws on data to identify the most vulnerable groups of people within the Rohingya and the host communities of Cox's Bazar, Bangladesh, during the COVID-19 pandemic. This study employed a systematic sequential method to identify the most vulnerable groups in the context of Rohingya and Host communities of Cox's Bazar. We conducted a rapid literature review (n = 14 articles) to list down Most vulnerable groups (MVGs) in the studied contexts during the COVID-19 pandemic and conducted four (04) group sessions with humanitarian providers and relevant stakeholders in a research design workshop to refine the list. We also conducted field visits to both communities and interviewed community people using In-depth interviews (n = 16), Key-informant Interviews (n = 8), and several informal discussions to identify the most vulnerable groups within them and their social drivers of vulnerabilities. Based on the feedback received from the community, we finalized our MVGs criteria. The data collection commenced from November 2020 to March 2021. Informed consent was sought from all participants, and ethical clearance for this study was obtained from the IRB of BRAC JPGSPH. The most vulnerable groups identified in this study were: single female household heads, pregnant and lactating mothers, persons with disability, older adults, and adolescents. Our analysis also found some factors that may determine the different levels of vulnerabilities and risks faced by some groups more than others in the Rohingya and host communities during the pandemic. Some of these factors include economic constraints, gender norms, food security, social safety-security, psychosocial well-being, access to healthcare services, mobility, dependency, and a sudden halt in education. One of the most significant impacts of COVID-19 was the loss of earning sources, especially for the already economically vulnerable; this

transcripts, and these are presented within the paper. All qualitative full transcripts are unavailable to the public because they contain personal identifying information about Rohingya and host participants. Ethical approval for the conduct of the study required that all data, including the locations of Rohingya participant camps, be de-identified, according to the data sharing policy of the Institutional Review Board (IRB) of BRAC University – an independent research ethics committee under BRAC University (https://bracpgsph.org/research-irb). For data access, qualified researchers may request de-identified data by emailing IRB of BRAC University at irb-jpgsph@bracu.ac.bd. The IRB is located at BRAC James P Grant School of Public Health, BRAC University 6th Floor, Medona Tower, 28 Mohakhali Commercial Area, Bir Uttom A K Khandakar Road, Dhaka-1213, Bangladesh.

**Funding:** This study was supported by the BRAC James P Grant School of Public Health, BRAC University. No additional external funding was received for this study. The funders had no role in study design, data collection and analysis, the decision to publish, or preparation of the manuscript.

**Competing interests:** The authors have declared that no competing interests exist.

had far-reaching consequences on individuals' food security and food consumption. Across the communities, it was found that the economically most affected group was single female household heads. The elderly and pregnant and lactating mothers face challenges seeking health services due to their restricted mobility and dependency on other family members. Persons living with disabilities from both contexts reported feelings of inadequacy in their families, exacerbated during the pandemic. Additionally, the shutdown in the formal education, and informal learning centres in both communities had the most significant impact on the adolescents during the COVID-19 lockdown. This study identifies the most vulnerable groups and their vulnerabilities amid the COVID-19 pandemic in the Rohingya and Host communities of Cox's Bazar. The reasons behind their vulnerabilities are intersectional and represent deeply embedded patriarchal norms that exist in both communities. The findings are essential for the humanitarian aid agencies and policymakers for evidence-based decision-making and service provisions for addressing the vulnerabilities of the most vulnerable groups.

## Introduction

The Rohingya refugees, also referred to as the forcibly displaced Myanmar nationals (FDMN) [1], are the world's largest internationally displaced population that reside in 34 camps in the Teknaf and Ukhiya sub-districts of Cox's Bazar, Bangladesh [2]. These camps are congested, and people live in cramped conditions, with five-to-six members living in single-room shelters [3, 4]. The terrains are hilly and often at risk of fires, floods, and mudslides which puts the entire population at risk [5]. The Bangladeshi host population lives alongside the Rohingya. It is one of the poorest population groups in the country, with around one-third of them living below the poverty line [6]. Before the arrival of Rohingyas, this population was economically vulnerable due to the scarcity of resources in the area. Since the last influx in 2017, the Bangladeshi host community has become especially vulnerable due to constant competition for scarce opportunities between them and the refugees.

Evidence has shown that the labor market has become competitive with Rohingya laborers entering the job market, willing to settle for lower wages than the local Bangladeshis and without the necessary documents [7]. Living expense has also escalated due to the increased demand for daily essentials [7, 8]. Moreover, the general perception of the host community stands that they have been neglected as they are not entitled to the humanitarian aid allotted only to the Rohingya community [8]. Furthermore, they are vulnerable to food shortages, inability to meet rent dues timely, and debt from health care expenses. Already in the low-income tier, they have little to no savings and are poorly equipped to cope with any drastic financial setbacks [8], leading to increased debts pushing them towards a vicious cycle of inter-generational poverty.

However, the hosts have better access to water and sanitation facilities such as latrines within their residence and community and home-based water sources, e.g., Tubewell. On the contrary, only half of the Rohingya population have access to sufficient water supply, necessary handwashing stations, and other WASH necessities. All water and hygiene facilities are in overcrowded areas and are meant to be shared by the households, i.e., communal facilities [9]. Due to these disproportionate services and facilities, there are rising tensions between the two communities [8] which the COVID-19 pandemic has further exacerbated.

The COVID-19 pandemic has taken a heavy toll on the largest refugee camps in Cox's Bazar, inflicting vulnerabilities and suffering on the two distinct communities thriving in these

areas. The Rohingya community has been severely affected by disruptions in services [10], the long waiting period in the health facilities [11], delays and reduction in the relief distribution (often claimed to have been disproportionate [12], closure of learning centres [12], and restricted mobility [10]. At the same time, the host communities have been adversely affected by the subsequent impacts of the COVID-19 pandemic, such as loss of livelihood opportunities [13], food insecurity [14], closure of health facilities and educational institutes [15], lack of transportation, and severe strains on household-level expenses, and mental health impact [16], which heightened the pre-existing vulnerabilities within both the Rohingya and host communities.

Though the impacts of Covid-19 have been well documented in the literature, the sub-groups of people rendered most vulnerable and marginalized during the pandemic and the impact of the COVID-19 pandemic on them have not been studied comprehensively. In any humanitarian crisis, the intersectional nature of the vulnerabilities faced by the beneficiaries is often ignored while planning responses [17, 18]. Hence, the interventions are accessible to the people who meet the most common vulnerability criteria that implementers know of and thus do not penetrate the deep hidden pockets of communities. As a result, many populations remain underserved and thus become 'most vulnerable' or often termed "ultra-vulnerable". For instance, women and girls in the camps are more vulnerable to Intimate Partner Violence (IPV) and Gender-Based Violence (GBV) [19] as polygamy, child marriage, dowry, separation, and abandonment are culturally accepted practices among the Rohingya [19], and this is compounded with the lack of street illumination at night, consequently, poses significant safety risks in and out of their homes coupled with mental health issues and lack of dignity and respect [20]. In the adjacent host community, there is a growing perception that the practices of polygamy and child marriages common in the Rohingya community may find their way to the host community, leading to further tension among the host community people and amplifying their effect on their worries [21]. Moreover, during the Covid-19, the women and girls face an increase in unpaid caregiving work [22] as more family members stay at home due to the lockdown in both sites [20], affected them greatly.

While the inclusivity of responses geared towards pandemic preparedness is imperative, identifying most affected groups, studying, analyzing, and considering their vulnerabilities in planning and designing such responses is crucial. However, identifying such groups continues to be a conceptual and methodological challenges because the extent and impact of vulnerabilities differ from individual to community level. Hence, this paper includes a participatory and inclusive approach to identify the most vulnerable groups within the Rohingya and the host communities of Cox's Bazar, Bangladesh, during the COVID-19 pandemic. By doing so, this paper will provide specific evidence to influence the interventions and policies to undertake the MVGs specific targeted approach. More specifically, this paper will help the policymakers and the implementers who will be working in such emergency situation by providing more contextual recommendations with all the nuanced and deeper analysis of the situations.

## Materials and methods

### Study design

This study employed a systematic sequential method to identify the sub-groups of populations deemed most vulnerable by the COVID-19 pandemic in the Rohingya camps and adjoining Host communities of Cox's Bazar Bangladesh. It presents partial findings from formative mixed-method research [23] conducted in ten Rohingya camps and four wards of the adjacent host communities in the Ukhiya sub-district of Cox's Bazar, Bangladesh, from November 2020 to March 2021.

**Table 1. Number of MVGs identified through three consecutive methods.**

| Methods | Rapid Literature Review | Research Design Workshop | Community Field Visit & Informal Discussion | |
|---|---|---|---|---|
| | | | Rohingya Community | Host Community |
| Types of MVGs | Women and children | Single female household heads | Women without husbands (widowed. abandoned or divorced) | Women- especially widows and separated with no formal education |
| | People with medical conditions | Persons with disabilities | Women who do not have children, especially male child | Women |
| | Elderly | Elderly | Families without male members | Day laborers |
| | Adolescents | Adolescents | Families having five to six female members | Adolescents |
| | Young married girls | People with chronic medical conditions | Families who solely rely on ration, especially when headed by women | Elderly people |
| | People with pre-existing respiratory infections | Pregnant and Lactating mothers | Families with adolescent children | Persons with disabilities |
| | Service providers | People who had lost livelihoods | Families with an elderly parent | Pregnant and lactating mothers |
| | | | Persons with disabilities | |
| | | | Pregnant and lactating mothers | |
| Total MVGs Identified | 7 | 7 | 9 | 7 |

In the first step, from August 2020 to October 2020, we listed down the sub-groups of populations in the studied contexts deemed most vulnerable by the COVID-19 pandemic through rapid literature review. In the second step, in November 2020, we conducted a research design workshop wherein the list was shared with the key local stakeholders for validating and updating the list based on stakeholders' experience and expertise of working in studied context. We refined our list of MVGs and finalized the selection criteria based on the feedback and inputs from the experts. In the third step, in November 2020, we conducted field visits in both communities, as it was crucial to assess from a community-centered perspective. We conducted interviews with community people to identify the most vulnerable groups and community-centered perspective regarding vulnerability and its social drivers. Table 1 presents the number of MVGs identified through these methods. Based on the data found through these field visits, we triangulated the final list of the MVGs. These methods are discussed in further detail below.

## Rapid literature review

We conducted a rapid literature review of the published academic papers, research, and program reports, policy and summary briefs, web-based articles from humanitarian websites, and newspaper articles relevant to our study area. The literature review aimed to understand and list down the sub-group of populations considered most vulnerable in the context of Rohingya camps and the host communities. The articles were searched-on and downloaded from the Google search engine and Google scholar using specific binding terms tabulated in Table 2.

**Table 2. Keywords used to scope published literature from March 2020 to September 2020.**

| Vulnerable (Combined by 'OR') | Host (Combined by 'OR') | Rohingya (Combined by 'OR') | COVID-19 (Combined by 'OR') |
|---|---|---|---|
| (a) | (b) | (c) | (d) |
| Marginalized | Local | Refugees | Coronavirus |
| At-risk | Resident | FDMN | |
| | Bangladeshi | | |

**Note:** 'a', 'b,' 'c,' & 'd' were combined using the Boolean operator 'AND.'

The search strategy was kept time-bound (March 2020 to September 2020) as we only intended to review the literature published since the onset of the COVID-19 pandemic in Bangladesh in March 2020. The articles had to be open-access, within our time frame, in English, focusing on the COVID-19 pandemic and the humanitarian crises in Cox's Bazar. Some articles, mainly research and program reports relevant to our objective, were also included in the database of our institution. A total of 14 articles [12, 20, 24–35] were analyzed to identify specific groups of populations from the Rohingya and the host communities of Cox's Bazar that were deemed most vulnerable during the COVID-19 pandemic. The literature review also aided in developing preliminary tools for the next two steps.

## Research design workshop

The expert consultation sessions were conducted during a Research Design Workshop (RDW), organized as part of the larger research project based on this study [23]. A total of 35 participants attended the workshop, including government representatives, non-government officials, and researchers working in our studied contexts. Participants were divided into four groups, and two researchers joined each group to assist the facilitation of the sessions. Four group sessions were conducted with participants wherein the initial list of MVGs and our preliminary research tools developed based on the findings from the literature review were shared with them. The participants reviewed those, and provided specific feedback for strengthening further. They refined the list and also provided the feedback on the framing of questions given the context and local terms to explain key concepts to the community people.

## Community field visits

The field visits were conducted to both studied areas in December 2020 and January 2021, respectively. The purpose of this visits was to validate the MVGs based on the ground realities, as they were theoretically defined from the literature review but identified more concretely through the research design workshop. Another objective of this field visits was to get a better idea about the community dynamics, their socio cultural and gender norms and practices.

During field visits, qualitative methods were applied for data collection. A total of 24 participants (12 in each community) were interviewed using in-depth (n = 16) and key-informant interviews (n = 8). To minimize the bias and obtain a representative sample, participants inclusion criterion was defined prior to the data collection. This criterion included the participants-1) belonged to study site 2) local resident of host community and living in the Rohingya camps 3) aged 30–65 years in case of adults 4) motivated to participate 5) belonged to the category of MVGs initially listed based on the literature review and workshop.

The IDIs were conducted with adult males and females, adolescent boys and girls and the KIIs were conducted with key community members including local elected members, community leaders, Camp in charge (CIC), and community volunteers or community health workers. Several informal discussions were conducted as well with the MVGs identified initially to better understand their situation. The participants were recruited using convenience and snowball sampling methods whereas two adult participants, each for an IDI and KII, was approached face-to-face first and then were asked to suggest names of other potential participants and so on until reached the point of redundancy. All the participants agreed to participate in the interview after we approached them.

The IDI and KII guidelines were pre-developed, pretested, and finalized beforehand. All the interviews, and also the informal discussions were conducted by six trained qualitative researchers, consisted of 4 males and 2 females (RS, AAP, MRH, ASMN, AJT, ZMI). All the researchers were public health professionals with extensive experience in qualitative research

methodologies including conducting IDIs, KIIs, and informal discussions. 12 experienced and trained local enumerators (4 males and 8 females)- well-versed with the local context and fluent in language of both communities, acted as interpreters during the interviews.

All the interviews were conducted either in Bangla (the language of the host communities) or in Ruáingga (the language of the Rohingya communities) and field notes were also taken during the interviews in Bangla. All the interviews- average 50 minutes long, were audio-recorded, and the recorded data were transcribed verbatim in Bangla within 24 hours by the local enumerators to reduce the data loss and then translated into English by a group of professional translators. None of the transcripts were returned to the participants since most of them were unable to read and write in English, or Bangla. The researchers cross-checked the data and familiarized themselves with it. Afterwards, inductive coding was done by the six researchers to identify key themes and sub-themes. Thematic analysis was conducted using Atlas Ti, qualitative software, version 9.0.

### Ethical considerations

The study was ethically approved by the Institutional Review Board (IRB) of the BRAC JPGSPH [36] under reference number IRB-6 November 20–057. Verbal informed consents were obtained from IDI and KII participants because of the contextual and local political sensitivity of the Rohingya issue in Bangladesh; informed verbal Assents were taken from the adolescent participants after consent was taken either from their guardian(s) or parent(s). Participants' privacy and data confidentiality were maintained at all the data collection, storage, and analysis stages. Each interview was conducted in a private place (mostly inside their house) selected by the participant. None of the non-participants was present during the interviews since the researchers politely diverted the curious community bystanders, and sometimes the enumerators conducted diversion interviews to keep them busy. Participant names were replaced with pseudonyms, and only limited personal information important for analysis was collected e.g., age, occupation, and education. All data was stored in a secured locker at the home institute of the researcher, where only the selected research team members had access. Furthermore, since this research was conducted during a pandemic, the research team followed recommended COVID-19 safety precautions during fieldwork to prevent coronavirus transmission to community members, participants, and researchers. All the IDI and KII participants were given hygiene packets (face masks, soap, toothpaste, toothbrush shampoo) as a token of appreciation for their participation in the study.

## Results

### Profile of the participants

Among the 24 participants, eight were adolescents aged 13–18 years, eight were adults aged 25–45 years, four were community leaders aged 39–45 years, and four were community health workers and volunteers aged 24–39 years. The Rohingya adolescent participants, both boys and girls were not studying during data collection while the host adolescent were all students. Furthermore, the adult men from both communities were involved in income generating activities while none of the adult women were in livelihood activities except the two health workers from the host and two community volunteers from the camp.

### Communities' perception on vulnerability

This paper explored communities' perception of who they perceived as vulnerable and the primary reasons behind their vulnerabilities (Table 3). According to our analysis some factors

**Table 3. MVGs among Rohingya and host communities and the reasons behind their vulnerabilities.**

| Rohingya Community | Host Community |
|---|---|
| **Women without husbands (widowed, abandoned or divorced)**- *are vulnerable arising from lack of safety and social security. They tend to be continually harassed and fear sexual and other form threats from men in the community.*<br><br>**Women who do not have children, especially male child**- *tend to be more vulnerable, are socially stigmatized. As sons are usually responsible for looking after their families*<br><br>**Families without male members**- *include those without income, an increased risk of insecurity; fear of thefts, violence and robberies. The female members are at greater risk of sexual abuse, sexual assault and rape by men in the community. Girl's safety is an issue for families concerned with preserving their family's honor and reputation*<br><br>**Families having five to six female members**- *are considered vulnerable due to cultural restriction, women are not allowed to earn nor work outside their homes. As a result, they are dependent on their families. Laws also favor males in terms of property rights and they have greater economic opportunities in general. Males are generally viewed as breadwinners while women are seen mainly as caregivers at home.*<br><br>**Families who solely rely on ration, especially when headed by women**- *are vulnerable as they don't have any other income source and ration are not always enough. They cannot eat as they wish.*<br><br>**Families with adolescent children**- *Daughters do not contribute financially to the family, but are dependent on their families for all their needs thus, often viewed as a liability, thus pushed to marry off at an early age. However, this 'solution' also burdens the families due to the existing dowry system. The situation worsened for families who were already struggling during the pandemic. In the case for boys, the sudden pause in learning and work opportunities meant they were likely to indulge in crimes and drugs.*<br><br>**Families with an elderly parent**- *are viewed as a burden as they are dependent on their family and unable to contribute.*<br><br>**Persons with disabilities**- *"Haat bhanga" (broken hands), "Lula" (one legged), are vulnerable groups as they are unable to earn their own livelihood and are economically dependent on others. Some of them also require assistance for eating and defacation (referring to the bedridden)."*<br><br>**Pregnant and lactating mothers**- *are vulnerable because they require high nutrition for the development of healthy baby, and additional care. Especially pregnant mothers can be physically constrained and at the later stages of the pregnancy are dependent on others for support.* | **Women- especially widows and separated with no formal education**- *are vulnerable as they are less likely to maintain their own livelihoods. Lack of education meant they could only work as day laborers and due to COVID-19 there were less or almost no opportunities for manual labor in the community thus making them economically vulnerable. Additionally, in the case of widows or separated women, they had no husbands to provide livelihood for them*<br><br>**Women**- *were identified as vulnerable because they were unable to work and earn as much as men which made them dependent on the male members of the household. Additionally, there are social and cultural restrictions on women's mobility as they are restricted from leaving the house, thus adding to their emotional and mental turmoil.*<br><br>**Day laborers**- *were deemed most-vulnerable due to their poor economic status as they are daily wage earners. Their income depended on availability. During lockdown, these opportunities no longer existed or have been stopped by the government. This means no-income for them resulting in exhausting savings in many cases.*<br><br>**Adolescents**- *are vulnerable as their education was hampered and had not resumed since the beginning of the pandemic.*<br><br>**Elderly people**- *identified as vulnerable because of their declining mental and physical capacity, many of them were dependent on others for livelihood and caregiving support. Thus, families view them as a burden.*<br><br>**Persons with disabilities**- *were identified as vulnerable due to their physical limitations that hindered them from working at par with other people without these limitations. Most of them were viewed as burden due to the dependency for their livelihood, and caregiving support. Moreover, their physical limitations also hindered them from getting married. Thus, impacting their mental health.*<br><br>**Pregnant and lactating mothers**- *are more at risk as they may face difficulties in accessing health care, and vaccination given the Covid-19 situation. They require additional care and medical support.* |

that may determine the different levels of vulnerabilities and risks faced by some groups more than others in the Rohingya and host communities during the pandemic included: economic constraints, gender norms, food security, social safety-security, psychosocial well-being, access to healthcare services, mobility, dependency, and a sudden halt in education. Based on these factors, the most vulnerable groups identified in Rohingya's context included: women without husbands, women who do not have children, especially male child, families without male members, families having five to six female members, families who solely rely on ration,

especially when headed by women, families with adolescent children, families with an elderly parent, persons with disabilities and pregnant and lactating mothers. Whereas in the host's context these included: women- especially widows and separated with no formal education, women, day laborers, adolescents, elderly people, persons with disabilities and pregnant and lactating mothers.

## Triangulated list of MVGs

Our findings explored a varied and diverse range of MVGs in Rohingya and host context. However, we identified five vulnerable groups common in both contexts (Table 4). In our study we identified these groups as the most vulnerable in the humanitarian context during Covid-19. Our informal discussions with the MVGs listed below explored how covid-19 further worsened their vulnerabilities.

**1. Single Female Household Heads (SFHHs).** As per our analysis SFHHs were vulnerable due to the absence of male members in the household which usually meant lack of safety and social security and little to no income.

According to our data, SFHHs were vulnerable to GVB and quid-pro-quo harassment. As in the context of the Rohingyas, the lack of a male members in the household meant the safety and security is compromised. Female participants mentioned that in some cases women in the Rohingya camps who have been estranged from their spouse faced significant insecurities in regards to thievery, robberies, and sexual violence including abuse, assault, harassment, and rape by men from both the Rohingya and host communities. There is a general concern regarding the security of the girls arising from the harassment, which tends to be a significant concern to the honor and reputation of the families involved. An adult female shared,

> *"My next-door neighbor's husband left her a few years back. She is only a girl. She could not go to the Bazar alone. Once she went to the pharmacy alone and returned home with tears in her eyes. Some men abused her verbally. They made some dirty jokes that she could not even utter. I do not know what happened there as she did not tell me in detail. But, since then, my husband does bazaar for her family"* (Adult Female, Rohingya).

On the other hand, women who are estranged lose their honor and social "status" in the host community. Where a husband was viewed as social 'protector' for 'social status and acceptance'. Being alone or having marriage breakdown led to stigma and gossip within the community and SFHHs faced the brunt of it. As one single woman mentioned that since her husband abandoned her, she was getting undesirable proposals (indicating sexual proposal), even from her relatives. Her husband was physically abusive yet it was better than what she faced now, she said, *"There is no respect for a divorcee woman within the community and I alone have to deal with the lustful approaches from men and gossips in the community". (SFHH, Rohingya)*

**Table 4. Final list of MVGs.**

| MVGs | Description |
|---|---|
| Single Female Household heads (SFHHs) without income/ low income | Women who are widow/divorced/abandoned by their spouses |
| People with Disabilities (PWDs) | Person with any type of disability and age 18+ years |
| Elderly | Both male and female of age 65+ years |
| Pregnant and Lactating Mothers (PLMs) | Currently pregnant or lactating mother with a child <2 years |
| Adolescents | Both boys and girls with age 10–19 years |

Being in an already vulnerable state, vulnerabilities of SFHH further increased during the pandemic. In both the Rohingya and host communities, they were considered the poorest groups as they lacked access to income earning opportunities as a result of the inbuilt and rigid gender and religious norms that restricted women's mobility within household boundaries. On the contrary, males–adult men, adolescent boys, and in some cases, older males, were expected to be the primary breadwinners of the families. In this aspect, SFHHs, mostly the households without male members meant that the absent of breadwinners, resulted in extreme economic hardship, led to severe food insecurity.

For instance, In the Rohingya camps, the majority of the participants mentioned that SFHHs- with little to no income, did not have any male members- primary providers for livelihood, thus were entirely dependent on the provided rations. Rohingya families were entitled to monthly rations (rice, dal, oil, chilies, onions, potatoes, and cash handouts that they get from NGOs/governments on a routine basis). During the pandemic, their hardships escalated given the disruptions, delays, and sometimes decreased in the rations. Moreover, the food which were sometimes perished when collected, rendered them inconsumable by the family members. As SFHHs without no income had insufficient food arising from this reduction. While male headed household could buy food with their income, SFHHs, however, could not. For example, one participant, whose husband died before the pandemic, mentioned that she craved for fish and meat but did not receive those in the rations provided during the pandemic. She lamented

"*We are worse off than the prisoners. Even in jail, prisoners get to have meat and fishes once a week. Since the pandemic begun, we could not even acquire those once a month*" (SFHH, Rohingya).

Another single mother mentioned that she even had to skip one meal daily to survive during the lockdown. In her own words,

"*As a single mother, I have always been suffering as I do not work nor earn, during the lockdown, the rations provided were barely sufficient, thus I had to skip one meal each day. If my husband is alive, he would have provided*" (SFHH, Rohingya).

The SFHHs in the host community faced similar severity of food insecurities. This was due to the group having fewer work opportunities as a result of their rigid social and cultural norms- that are similar to the Rohingyas. However, SFHHs with at least secondary education, could work for NGOs and schools, but they were minute in number as most did not attain any formal education. The lack of training availability for them, meant the only options they had either [e.g., *Rasta r kaj kora*, *Rajmistri*, *Kamla*, *etc.* (day laborers)] or working as a domestic maid while men had greater access to income opportunities ranging from rickshaw pulling, to owning grocery shops and other informal business. However, the lockdown restricted the earning opportunities for this community, all types of work were affected, and SFHHs faced it's brunt as most of them lost their only means of earning which were already difficult to obtain. Thus, they were left with no livelihood opportunities during the pandemic whereas fears of starvation were uppermost on their minds.

As instance, one SFHH mentioned that before the pandemic, she worked as a housemaid which came to an abrupt end when the lockdown was announced as her employer prevented her from coming to work due to the fear of catching Covid-19. She could not manage any other job due to the lack of required skill and education. Moreover, as a low-paid worker, she barely had any savings for herself and her family. Thus, she had to borrow money and food from neighbors. Furthermore, the chances of getting another job during the lockdown was

almost impossible which meant her situation would only worsen. Therefore, her family had been surviving the pandemic by reducing their consumption to one meal a day, sometimes starving for days. She narrated,

> *"I lost my only income source during the pandemic, and now have been struggling to manage any other job. Since the pandemic started my family had been surviving by the donation of the neighbors. But how long am I going to get that support? I have been struggling to arrange only two meals for my family, and sometimes had to remain hungry for days"* (SFHH, Host).

The combination of these factors such as the increased risk of insecurity and harassment, severe food insecurity issues have pushed the SFHHs from both communities to live an extremely vulnerable life during the pandemic.

**2. Persons with Disabilities (PWDs).** As per our analysis, PWDs were vulnerable due to their "*Durbolota*" (physical conditions and limitations) such as being unable to walk, speak, hear, and work effectively as others, on top of being in a constant need for caregiving support.

Our data found that the physical condition and limitations faced by PWDs makes them dependent to their caregivers. Some of them are in constant need for assistance to perform daily basic activities such as eating, bathing and using the toilet. Hence, PWDs were viewed as a burden, thus neglected by their own family members.

In both communities, caregivers of PWDs were usually their female family members. Furthermore, the Covid-19 pandemic had increased the caregiving responsibilities of females in the both communities suggesting that caregivers could not fully commit to the need of the PWDs, thus increasing their vulnerability during the pandemic.

One male PWD from the Rohingya camp, mentioned that he was unable to walk since birth, there was on constant need for a caregiver, his wife for all his daily basic activities, such as bathing and using toilet. His wife's duties increased during the pandemic and was unable to care for him as much as she used to. As he had a large family consisting of children and other members that are now constantly at home also required her to cater for them, therefore compromising her time previously allocated to him. Given his physical condition, and lack of ability to fend for himself, he faced further internalized stigma from being a burden. Thus, the pandemic had pushed him into the feelings of helplessness, therefore taking a further toll on his already poor mental state. He said,

> *"I require constant assistance as I have been unable to walk. i cannot even use the toilet without my wife's assistance, who now has more responsibilities. Therefore, if I have to use the toilet, I would wait until she is available to tend to me. Pandemic has made my life worse"* (Male PWD, Rohingya)

Likewise, one female PWD from the host community mentioned that he had suffered from internalized stigma affecting his morale. The participant shared that,

> "*We, the PWDs, are invisible to them. . .. Even in our own home, we were always neglected as we are dependent. . .. We need support from families, hospital staff, and community people. . .. During the pandemic, I needed to go to the hospital, but my caregiver was engaged with other matters. So, I had to suffer in silence. At the time I felt invisible to everyone around me. (Female PWD, Host).*

Furthermore, some participants from host community mentioned about the economic dependency of the PWDs. Besides being physically dependent, PWDs were also dependent on

their family members and other community members for their conditions that restricted them from work. In this aspect, this populations faced limitations in access to work of his and life and were viewed as invisible and excluded from society. However, some PWDs that were able to work such as operating grocery shops are also affected by the pandemic as they were not allowed to conduct their businesses during the lockdown. One PWD said,

> *"Because I am an amputee without one arm, I feel like I am less of a person as I cannot perform as efficiently as other people, therefore I opened my own small grocery shop so I could provide for myself and family, however when the lockdown happened, all my hopes for a better living were suddenly gone"* (Male PWD, Host).

**3. The elderly.**   The elderlies were viewed as equally vulnerable as PWDs due to their age, declining mental and physical capacity, inherent dependency, and need for regular medical attentions.

The majority of the participants from both the Rohingya and host communities mentioned that due to their age, and physical condition, the elderlies were dependent on others for care. Moreover, owing to the preexisting comorbidities, the elderlies need to visit hospitals, and health facilities on a regular basis, thereby they need someone to accompany them. In this aspect, the lack of caregivers in the household, and the unavailability of caregivers meant that their regular health care needs are sometimes ignored. This situation worsened during the pandemic. For instance, some elderlies from both the communities mentioned that ever since the lockdown began, they had difficulties visiting health facilities due to the lack of caregivers, who were already engaged with other duties at home. Moreover, the caregivers were reluctant to visit hospitals that were viewed as a COVID-19 infection hotspot. Thus, this prevented the elderlies from receiving their much-needed medical care. One elderly said,

> *"I am diagnosed with diabetics, high-blood pressure, asthma and need to visit hospital for frequent follow-ups. once i fell very ill during COVID, but none of my family members would take me there as they feared getting infected from hospital patients."* (Female Elderly, Rohingya)

Furthermore, health disruptions were faced by both communities. Some of the health facilities were closed during the lockdown as mentioned by the participants. Thus, some of the elderlies had to walk long distances to the next available facilities. This was laborious given their health conditions. Additionally, these health facilities were insufficiently equipped in providing the needed services. Some were even overcrowded with patients while the service providers and support persons were very few in numbers causing longer waiting lines.

For instance, one adolescent girl from the Rohingya camp mentioned that her elderly grandmother's health worsened while waiting at the long queue. During the lockdown, her grandmother desperately needed medical attention, they made the long walk to a distant health point as the nearest one abruptly closed. Additionally, she also had to stand in the long queue as hospitals were understaffed. This situation was quite new and unusual at the time. Her grandmother's condition worsened after the experience. She said,

> *"My grandmother got worse after walking a long hilly journey to the hospital. Moreover, the hospitals were crowded with patients resulting hours long queue. It was quite difficult for my grandmother to stand in queue given her health condition. After experiencing all that, her condition worsened"* (Adolescent girl, Rohingya)

It has been evident that disruption on existing health care services from covid-19 had a disproportionate impact on people across all age, in turn, no measures were undertaken for those affected.

This situation was worse in the host community. Compared to the Rohingya community, the host community only had a limited number of healthcare facilities during the pandemic. Therefore, the elderlies in the host communities faced more difficulties accessing health services as compared to elderlies from the Rohingya communities. Furthermore, elderlies from host community faced a heavy toll on their mental health as they were isolated in their homes due to the fear of contracting covid-19 given their age and preexisting comorbidities.

For instance, one elderly from the host community shared that she was terrified during the early days of Covid-19 since she heard that elderlies were at more risk of Coronavirus as death rates of elderlies were high. Her family was aware about this, and hence she was kept in isolation which meant she could not socialize, thus affecting her mental health. She said,

> "*I was very scared when this virus first was identified in Cox's bazaar. So, my family did everything they needed to do at the time. I spent all my time at home then, this made me very upset*" (Elderly Woman, Host)

This quote highlights that poor mental state of the elderlies during pandemic arising from the death consequences of Covid-19 associated with older age. Further increasing their vulnerability.

**4. Pregnant and Lactating Mothers (PLMs).** PLMs were considered vulnerable as they required additional care, medical support, and proper nutrition. According to our analysis, the potential determinants behind their vulnerabilities were: lack of care and assistance, inadequate healthcare services, nutritional deficiency, and poor mental health conditions. The presence of these determinants during the pregnancy, and/while breastfeeding have had serious consequences for the mothers, and the babies, hence PLMs living in humanitarian context were potentially at risk. Coupled with the detrimental impacts of Covid-19- loss of livelihood, and disruption in healthcare services, the vulnerability of the PLMs escalated in both studied contexts.

Our study found that Covid-19 had massively affected the livelihood opportunities, leading to severe food insecurity for Rohingya, and adjacent host communities. For the Rohingyas, the level of food insecurity was more severe than the host community. As in many respondents from the Rohingya community mentioned that the amount of ration they were provided in usual time was decreased in quality during the early lockdown period. In this context, ensuring proper nutrition for the PLMs, which was fundamental for the survival and well-being of the mother and the child -in the womb and throughout the early childhood, was next to impossible in many cases. Therefore, the PLMs were considered nutritionally at-risk.

As one female, mother of a 2-month-old daughter, shared that she had been suffering from severe weakness due to malnutrition during her pregnancy. During the early lockdown period, she received ration contained limited food items lacking in essential nutrient, which was insufficient enough to meet her daily nutritional requirements, making her vulnerable to malnutrition, and poor health condition. She said,

> "*When I was pregnant, I became so thin that my bones were visible*! *This was obvious as I did not eat fish nor meat then.*" (*Lactating Mother, Rohingya*).

Albeit, the food insecurity was comparatively less severe in the host community, nutritional deficiency was still prevalent among the PLMs, lack of care and assistance from the family

members further increased their vulnerabilities. According to the participants from the host community, during the early lockdown period while the male members were staying at the home, female members- despite of being pregnant, or lactating were tasked with looking after their male counterparts as per the gender roles within the community. Sometimes PLMs had to skip meals, to ensure that the male members had enough. Coupled with the increased work-load at home and little to no rest, skipping meals worsened their health condition, very detrimental for a healthy pregnancy and babies. One community health worker said,

> *"During the pandemic, the pregnant and lactating mothers were often at-risk due to their poor health condition. With this health condition, we cannot expect any good"* (Community health worker, Host)

Additionally, our data found a disruption in healthcare services in both studied contexts during the lockdown. According to some of our KII respondents, some of the medical facilities were closed during the lockdown, and the available health facilities were understaffed which created obstacles for the PLMs seeking healthcare services. In this aspect, the PLMs in the host community were more vulnerable than the PLMs in the Rohingya community since the host had comparatively less health facilities available during the pandemic. As one community health worker said,

> *"The available health facilities in the host community were fewer than the Rohingya community. The Rohingyas had comparatively more health posts, camp hospitals open during the lockdown. Therefore, PLMs from the host community faced more challenges to avail ANC and PNC services. Some of the hospitals were understaffed to handle and attend emergency cases. Therefore, many of the pregnant mothers from host community had to endure a lot during their delivery. Some lactating mothers could not access the vaccination required for their babies"* (Community health worker, Host)

Owing to inadequate healthcare services, the possible obstacle and challenges PLMs faced during the lockdown period was quite evident from this quote. This situation led to uncertainty of getting services on time, further increased anxiety among the PLMs. This indicated the poor mental health condition of the PLMs, making them more vulnerable to poor pregnancy and neonatal outcomes.

**5. Adolescents.** Our analysis suggests that, adolescent girls were considered vulnerable to child marriage and poor mental health condition. Whereas adolescent boys were considered most vulnerable because of the community feared drug abuse, and criminal activities and hampering of educational prospects.

In the Rohingya community, daughters do not contribute financially to the family, but are dependent on their families for all their needs thus, often viewed as a liability. Therefore, they are pushed to marry off at an early age. However, this 'solution' also burdens the families' finances as marrying off the girls entailed financial setbacks due to expectations and demands regarding dowry.

As one adult male shared that around BDT 2.5 Lakh (US$2,889 equivalent) could be cost for one wedding. Even for a groom who does not earn, his family is expected to receive goods or cash, which include *"Khaat Palong"* (bed and mattress), *"Chula"* (stove), and other household items from the bride's family. During the pandemic, meeting these expectations in the future, given the crisis, was seen as next to impossible and an added burden for the girls' families. Therefore, these families were not only grappling with uncertain times due to the pandemic but also remained anxious about what they would do about their adolescent girls in the future.

On the other hand, girls from host community were vulnerable amid the pandemic because their education was hampered and had not resumed since. According to many adolescent participants, educational institutions' sudden and lengthy closure has hampered their educational prospects, leading to a precarious future. As in the case of one adolescent girl who identified herself as vulnerable because she could not go to school, meet her friends, and socialize, which added to her emotional turmoil.

Another adolescent girl also shared similar experiences. However, in her case, she went through extreme mental stress as she was about to be married off at a very early age. The lockdown caused her father to lose his only source of income. Thus, her father decided to marry her off as the only possible option to relieve some of the financial burden of the family, being the sole provider of the household. This caused the girl to go through extreme anxiety as she believed it was in her best interest to continue her education, and now finds it unsettling whenever she thinks of her future. She lamented,

> "*I do not want to get married; rather, I want to pursue my studies further. But my father wants to marry me off as he has been struggling financially. I need help to continue my studies.*" (Adolescent girl, Host).

Though in the case of Rohingyas, there was only few that mentioned about the adolescent boys. As per them adolescent boys were likely to indulge in crimes and drugs due to the sudden pause in learning and work opportunities. On the other hand, in the host community, some of the respondents considered adolescent boys as equally vulnerable as adolescent girls. They mentioned that adolescent boys also have to go through the similar uncertainty during the pandemic regarding their future educational prospects.

## Discussion

This study identifies five groups of population as the most vulnerable groups within the Rohingya and the host communities of Cox's Bazar, Bangladesh, during the pandemic. The findings of this study explored that the reasons behind these group of population's vulnerabilities are intersectional, due to that, these groups have had different levels of exposure during Covid-19 pandemic.

In our studied contexts, the SFHHs were vulnerable to gender and sexual violence due to the absence of social safety and security- associated with the notion of viewing males as the protector of females. Whereas families with several women members or several girls of marriageable age were comparatively less vulnerable in terms of social safety-security and economic well-being since those families may have male members. A qualitative study conducted among female-headed households in Kermanshah, West of Iran, in 2019 showed that the female-headed households were identified as the most vulnerable groups that may be abused and were subjected to violence by society and lack the security to live and maintain their families [37]. Coupled with the catastrophic impact of the COVID-19 pandemic, these households have been suffering from extreme economic hardships, leading to food insecurity wherein skipping meals was common to provide for their children as per our findings. Therefore, female led households found to be amongst the poorest household while many other studies also explored the similar hardships of these households, hence often regarded as the "poorest of the poor" [38–40]. Hindsight identifies the "feminization of poverty" [38]- the process whereby poverty becomes more concentrated among Individuals living in female lead households—as a critical concept for describing single-female-headed families at social and economic levels. As a report, published in the context of Afghanistan, claimed that Female-

headed households bear the brunt of Covid-19 as livelihood gaps increase. The report has dictated that the covid 19 crisis has significantly hampered involvement of women in economic activities. Primarily within the informal sector, where women makeup over half of the work force. The lockdown resulted in sudden travel bans leading to further widening of gender livelihood gaps, thus the worsening of food insecurity within the households [41].

The PLMs were considered as vulnerable because they required support and care for the proper development of a healthy baby. However, COVID-19 had massively affected the livelihood opportunities, food security, and healthcare services during the lockdown- indicated the lack of nutrition for the PLMs. Other studies and reports have also claimed similar findings [40]. The impact of COVID-19 exacerbated the levels of food insecurity in many marginalized contexts leading to the birth of malnourished babies [40, 42]. Moreover, COVID-19 lockdowns disrupted healthcare service delivery, affecting pregnant and lactating mothers' access to healthcare [40, 43] in humanitarian contexts.

Older Adults and Persons with Disabilities were deemed vulnerable due to their poor and weak health conditions. In our study, these two groups were regarded as the most susceptible due to their age, preexisting comorbidities, inherent dependency, and need for constant medical attention. Due to these reasons, they required caregivers who, in turn, were overburdened with increased household responsibilities during the lockdown and hence were not always able to respond to their needs. These findings have further reinforced the evidence that the disproportionate burden of COVID-19 related risks faced by those aged 60 and above, people afflicted with co-morbidities, disabilities, malnutrition, and other marginalized groups [44].

Recent studies from Bangladesh and India have highlighted that the availability of basic health care has been severely impacted during the pandemic [45, 46]. As per the evidence from our study, due to the disruptions in healthcare services during the pandemic, especially the older people suffered a lot to avail their regular healthcare needs, whilst the key factors affected them were; long distance to the facilities, long waiting hours and the less support persons in the hospitals. Evidence from other studies also depicted similar findings, for instance, a study showed that key factors impacting elderlies' access to healthcare services during the pandemic include long distance to the facilities, loneliness and perceived vulnerability of older people (Risk of COVID 19 and need for care). This study also stated that since thousands of older Rohingya adults experience communicable and non-communicable diseases, whilst limited medical services mean they are prone to a worse COVID 19 risk, resulted in having fewer medicines and obstacles to obtain their routine medical care. It is thus suggested that policy makers focus more on targeted interventions for elderlies' [47].

Our findings emphasized the increased vulnerabilities facing adolescent girls and boys in Rohingya camps and host communities in Cox's Bazar amid the COVID-19 pandemic. We highlight that adolescents are at more risk of lack of schooling, which may put the girls at more risk for early child marriage, leading poor mental health condition. Literature indicates that Child marriage is widely socially accepted and common despite being illegal under Bangladesh law [40, 48, 49]. The dowry system and child marriage have been a tradition enforced by the Rohingya community in the past and have continued to the camp settlements in Cox's Bazar [19]. It is reported that the dowry system ranks amongst camp residents' deepest social justice concerns [50]. The dowry system is seen as a driver of child marriage as well [19].

This evidence intersects with literature reporting that health crises, coupled with educational and economic disruptions in humanitarian contexts [12], compounded adverse effects on adolescents and their ambitions and goals in life.

The study depicted how the identified most vulnerable groups affected by the Covid-19 pandemic. Since, the needs of the most vulnerable groups varies from each other, it was not possible to address all of their needs entirely with an intervention that was designed without

considering the intersecting nature of their vulnerabilities. Therefore, our study proposes to undertake a targeted approach when a crisis unfolds.

## Biases and limitations

Researcher's bias was one big bias of this study since the researchers came with pre-conceived notions and outside (etic) perspectives, leading to have a certain perspective towards the participants. In many cases, they may have led to targeted identification of certain groups as most vulnerable that the researchers deemed obvious. However, this bias was mitigated by continuous acknowledgment of researchers' reflexivity and positionality through group discussions within the team during the data collection and data analysis. Another bias related to this study was recall bias since this study was conducted after the lock down had been lifted off. Moreover, the researchers asked participants to retrospectively comment on lockdown times. This may have induced recall bias and affected the accuracy of the responses. However, this was mitigated by reminding the participants of major events happened during the lock down period in an incremental way.

One limitation of this study is that it was not conducted in all the Rohingya camps and adjoining host communities. Therefore, the results of this study may not be generalized across all Rohingya and host populations. However, the systematic sequential methods employed in study design makes this study robust and applicable to the sites it was conducted in.

## Conclusion

This study identifies five MVGs amid the COVID-19 pandemic in the Rohingya and Host communities of Cox's Bazar; Single female-headed households, pregnant and lactating mothers, persons with disability, older adults, and adolescents. Since the drivers of their vulnerabilities are intersectional, this study makes the following recommendations in the context of Covid-19 pandemic, considering their intersectional nature of vulnerabilities:

*First*, humanitarian providers in Cox's Bazar need to consider targeted food and monetary stipends in Rohingya camps and host communities for all the most vulnerable groups, particularly the female-headed households. *Second*, it is critical to provide focused skills development and training to female-headed families, including facilitating job placements for them in govt/ NGO and other agencies; the same applies to PWDs, wherever feasible. *Third*, there is a dire need to ensure sufficient human resource capacity in the health facilities of both studied areas. *Fourth*, campaigning on mental health issues exacerbated by COVID-19 and an increase in the provision of psychosocial services across both communities addressing all age groups, especially for the elderlies, and adolescents has become imperative. *Lastly*, the health care facilities need to introduce priority waiting areas, queues, and shorter waiting times for the elderly, pregnant women, and PWDs. The needs and priorities of these most vulnerable groups need to be incorporated into future programs and policies with stakeholders.

## Supporting information

**S1 Checklist. Completed COREQ checklist.**
(DOCX)

## Acknowledgments

The authors are also thankful to the larger research team members; Dr. M Shafiq Rahman, Kazi Sameen Nasar, Saifa Raz, Abdul Jabbar Topu, ASM Nadim, Zuhrat Mahfuza Inam, and Dr. Ashrafuzzaman Khan, for their support during the data collection.

## Author Contributions

**Conceptualization:** Rafia Sultana, Ateeb Ahmad Parray, Muhammad Riaz Hossain, Bachera Aktar, Sabina Faiz Rashid.

**Data curation:** Rafia Sultana, Ateeb Ahmad Parray, Muhammad Riaz Hossain.

**Formal analysis:** Rafia Sultana, Ateeb Ahmad Parray, Muhammad Riaz Hossain, Bachera Aktar, Sabina Faiz Rashid.

**Funding acquisition:** Bachera Aktar, Sabina Faiz Rashid.

**Investigation:** Rafia Sultana, Ateeb Ahmad Parray, Muhammad Riaz Hossain, Bachera Aktar.

**Methodology:** Rafia Sultana, Ateeb Ahmad Parray, Muhammad Riaz Hossain, Bachera Aktar, Sabina Faiz Rashid.

**Project administration:** Ateeb Ahmad Parray, Muhammad Riaz Hossain, Bachera Aktar.

**Resources:** Rafia Sultana, Bachera Aktar.

**Software:** Muhammad Riaz Hossain, Bachera Aktar.

**Supervision:** Rafia Sultana, Ateeb Ahmad Parray, Muhammad Riaz Hossain, Bachera Aktar, Sabina Faiz Rashid.

**Validation:** Rafia Sultana, Ateeb Ahmad Parray, Muhammad Riaz Hossain.

**Visualization:** Rafia Sultana, Muhammad Riaz Hossain.

**Writing – original draft:** Rafia Sultana.

**Writing – review & editing:** Rafia Sultana, Ateeb Ahmad Parray, Muhammad Riaz Hossain, Bachera Aktar, Sabina Faiz Rashid.

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
