## [Decision Letter · Decision Letter 0]

14 Jul 2022

PGPH-D-22-00606

“We are invisible to them” - Identifying the most vulnerable groups in humanitarian crises during the COVID-19 pandemic: The Case of Rohingyas and the Host communities of Cox’s Bazar

Dear Dr. Sultana,

Thank you for submitting your manuscript to PLOS Global Public Health. After careful consideration, we feel that it has merit but does not fully meet PLOS Global Public Health’s publication criteria as it currently stands. Therefore, we invite you to submit a revised version of the manuscript that addresses the points raised during the review process.

We look forward to receiving your revised manuscript.

Kind regards,

Christian Wejse, MD, PhD, Assoc.Prof

Academic Editor

Journal Requirements:

1. Please amend your detailed online Financial Disclosure statement. This is published with the article. It must therefore be completed in full sentences and contain the exact wording you wish to be published.

2. In the Funding Information you indicated that no funding was received. Please revise the Funding Information field to reflect funding received.

Please ensure that the funders and grant numbers match between the Financial Disclosure field and the Funding Information tab in your submission form. Note that the funders must be provided in the same order in both places as well.

3. Please update your online Competing Interests statement. If you have no competing interests to declare, please state: “The authors have declared that no competing interests exist.”

4. Please provide separate figure files in .tif or .eps format and remove any figures embedded in your manuscript file. Please also ensure that all files are under our size limit of 10MB.

Additional Editor Comments (if provided):

Reviewers' comments:

Reviewer's Responses to Questions

**Comments to the Author**

1. Does this manuscript meet PLOS Global Public Health’s publication criteria? Is the manuscript technically sound, and do the data support the conclusions? The manuscript must describe methodologically and ethically rigorous research with conclusions that are appropriately drawn based on the data presented.

Reviewer #1: Yes

Reviewer #2: Yes

2. Has the statistical analysis been performed appropriately and rigorously?

Reviewer #1: N/A

Reviewer #2: N/A

3. Have the authors made all data underlying the findings in their manuscript fully available (please refer to the Data Availability Statement at the start of the manuscript PDF file)?

Reviewer #1: Yes

Reviewer #2: No

4. Is the manuscript presented in an intelligible fashion and written in standard English?

Reviewer #1: Yes

Reviewer #2: Yes

5. Review Comments to the Author

Reviewer #1: • From what I understand, this paper sets out to identify the most vulnerable groups within both the Rohingya and host communities of cox bazaar. It does so using 3 separate pieces of information. It identifies many groups as vulnerable.

• However, what I feel the study fails to do is accomplish its main task – that is to identify THE MOST vulnerable group. Instead, it describes many forms of vulnerability that as a result affect a large swath of the population. COVID-19 is an additional element in the study but its role further complicates rather than focuses the discussion around vulnerability. As such, given the findings are so broad, I don’t know what I learn from the study.

• The study makes recommendations but I felt these were premature without a more detailed discussion around what it means to be most vulnerable and thus how one would want to prioritise policy attention.

• I think there is excellent data underlying this paper but without more focus, it is too long and does not make a central argument or present really insightful findings. The authors could revise the paper with a more focused approach and make a more meaningful contribution to the literature.

Reviewer #2: This article aims to identify the most vulnerable populations within the Rohingya refugees and those living in host communities nearby Rohingya refugee camps during the COVID-19 pandemic. This paper uses riche qualitative data from published literature, focus group discussions, in depth interviews, and key informant surveys to identify the most vulnerable populations.

While this paper provides rich information on these populations and their experiences during the COVID-19 pandemic, the paper could be significantly streamlined and clarified in terms of its contribution to the literature. I list a few major and minor comments below that I intend to be helpful in improving the manuscript.

1. The introduction sets the article up as a comparison between the host communities and the Rohingya refugees, but the analysis itself and resulting discussion is not situated as a comparison between the two populations -- i.e., presenting and discussing why particular subgroups are more vulnerable in one setting than the other, highlighting cases where populations are vulnerable in both settings. Alternatively, focus in on one population alone (e.g., Rohingya). As the paper is currently written, the analysis and discussion does not speak to the introduction.

I also think the introduction needs a paragraph outlining what the study is, how it was implemented, and what it is contributing to the literature.

2. The introduction needs to be significantly streamlined and updated. there seems to be some literature missing in the introduction that could help fram the article and better situate the contribution within the literature. For example:

Barua A, Karia RH. Challenges Faced by Rohingya Refugees in the COVID-19 Pandemic. Annals of Global Health. 2020; 86(1): 129, 1–3. DOI: https://doi.org/10.5334/aogh.3052

Gulglielmi, S., Seager, J., Mitu, K., Baird, S., Jones, N. Exploring the impacts of COVID-19 on Rohingya adolescents in Cox's Bazar: A mixed-methods study. Journal of Migration and Health. 2020; 1-2. https://doi.org/10.1016/j.jmh.2020.100031

Zakir Hossain, ANM. Sustainable Development and Livelihoods of Rohingya Refugees in Bangladesh: The Effects of COVID-19. International Journal of Sustainable Development and Planning. 2021; 16(6): 1141-1152.

The latter of the three could nicely highlight that policy responses are inadequate and this article contributes by identifying most at risk groups.

A few other places that need citations;

Sentence at line 89 "Evidence as shown...

Sentence at line 92 "Moroever, the general perception..."

Line 123 "Since the introduction of.." is there not anything more recent than July 2020 at this point?

3. In general the paper should be edited for clarity. In several locations there is awkward phrasing, e.g.:

-- In the abstract "Covid-19 pandemic has had generally an adverse impact..." I would either drop the generally or move it to be "has had a generally adverse impact"

-- first sentence of introduction. Should have commas around "also referred to...(FDMN). and "who reside" should be "that reside"

-- line 83 "On the other hand' -- there is not really a juxtaposition being presented here so strange to say.

-- Line 111 - suggest switching the 'on the other hand' to "at the same time"

Other notes:

In the abstract: Sentence starting "The finalized list of the most vulnerable groups..." the ; after methods should be a : l I would also use a : in the results section "identified in this study were: single

Table 1 is very difficult to read.

Suggest having a copy-editor go through the paper. E.g.,:

line 120, "escalated from the stigma" - delete the

line 140 "."r instance"

line 175 "local communities" Do you mean leaders?

6. PLOS authors have the option to publish the peer review history of their article (what does this mean?). If published, this will include your full peer review and any attached files.

**Do you want your identity to be public for this peer review?** For information about this choice, including consent withdrawal, please see our Privacy Policy.

Reviewer #1: No

Reviewer #2: No

---

## [Editor Report · Decision Letter 1]

2 Feb 2023

PGPH-D-22-00606R1

“We are invisible to them” - Identifying the most vulnerable groups in humanitarian crises during the COVID-19 pandemic: The Case of Rohingyas and the Host communities of Cox’s Bazar

Dear Dr. Sultana,

Thank you for submitting your manuscript to PLOS Global Public Health. After careful consideration, we feel that it has merit but does not fully meet PLOS Global Public Health’s publication criteria as it currently stands. Therefore, we invite you to submit a revised version of the manuscript that addresses the points raised during the review process.

PLOS Global Public Health considers qualitative and mixed-methods studies for publication. We recommend that authors use the COREQ checklist, or other relevant checklists listed by the Equator Network, such as the SRQR, to ensure complete reporting (http://journals.plos.org/globalpublichealth/s/submission-guidelines#loc-qualitative-research). 

In general, we would expect qualitative studies to include the following: 1) defined objectives or research questions; 2) description of the sampling strategy, including rationale for the recruitment method, participant inclusion/exclusion criteria and the number of participants recruited; 3) detailed reporting of the data collection procedures; 4) data analysis procedures described in sufficient detail to enable replication; 5) a discussion of potential sources of bias; and 6) a discussion of limitations. 

EDITOR: Please insert comments here and delete this placeholder text when finished. Be sure to:

Indicate which changes you require for acceptance versus which changes you recommendAddress any conflicts between the reviews so that it's clear which advice the authors should followProvide specific feedback from your evaluation of the manuscript

Please ensure that your decision is justified on PLOS Global Public Health’s publication criteria and not, for example, on novelty or perceived impact.

We look forward to receiving your revised manuscript.

Kind regards,

Christian Wejse, MD, PhD, Assoc.Prof

Academic Editor
---

## [Editor Report · Decision Letter 2]

21 Mar 2023

PGPH-D-22-00606R2

“We are invisible to them” - Identifying the most vulnerable groups in humanitarian crises during the COVID-19 pandemic: The Case of Rohingyas and the Host communities of Cox’s Bazar

Dear Dr. Sultana,

Thank you for submitting your manuscript to PLOS Global Public Health. After careful consideration, we feel that it has merit but does not fully meet PLOS Global Public Health’s publication criteria as it currently stands. Therefore, we invite you to submit a revised version of the manuscript that addresses all the points raised during the review process, and not only those you find suitable. A completed COREQ checklist will be a prerequisite for acceptance, otherwise the manuscript will be rejected for not living up to the PLOS GPH publication criteria.

A rebuttal letter that responds to each point raised by the editor and reviewer(s). You should upload this letter as a separate file labeled 'Response to Reviewers'.A marked-up copy of your manuscript that highlights changes made to the original version. You should upload this as a separate file labeled 'Revised Manuscript with Track Changes'.An unmarked version of your revised paper without tracked changes. You should upload this as a separate file labeled 'Manuscript'.A completed  COREQ or similar checklist

We look forward to receiving your revised manuscript.

Kind regards,

Christian Wejse, MD, PhD, Professor in Global Health

Academic Editor

Journal Requirements:

Additional Editor Comments (if provided):

The manuscript has only been revised according to some of the comments given. The response that some of the comments not are addressed because of the length of the manuscript, are not satisfactory, this is an online journal and page printing cost is not a relevant issue to raise. Therefore please address:

5) a discussion of potential sources of bias; and

6) a discussion of limitations.

Please also submit the completed COREQ checklist with the revised manuscript
---

## [Editor Report · Decision Letter 3]

2 May 2023

“We are invisible to them” - Identifying the most vulnerable groups in humanitarian crises during the COVID-19 pandemic: The Case of Rohingyas and the Host communities of Cox’s Bazar

PGPH-D-22-00606R3

Dear Ms Sultana,

We are pleased to inform you that your manuscript '“We are invisible to them” - Identifying the most vulnerable groups in humanitarian crises during the COVID-19 pandemic: The Case of Rohingyas and the Host communities of Cox’s Bazar' has been provisionally accepted for publication in PLOS Global Public Health.

Best regards,

Christian Wejse, MD, PhD, Professor in Global Health

Academic Editor